# Quantitative Assessment of Surge Capacity in Rwandan Trauma Hospitals: A Survey Using the 4S Framework

**DOI:** 10.3390/ijerph22101559

**Published:** 2025-10-13

**Authors:** Lotta Velin, Menelas Nkeshimana, Eric Twizeyimana, Didier Nsanzimfura, Andreas Wladis, Laura Pompermaier

**Affiliations:** 1Centre for Teaching & Research in Disaster Medicine and Traumatology (KMC), Department of Biomedical and Clinical Sciences, Linköping University, Johannes Magnus väg 11., 583 30 Linköping, Sweden; andreas.wladis@liu.se (A.W.); laura.pompermaier@liu.se (L.P.); 2Department of Internal Medicine, Centre Hospitalier Universitaire de Kigali, KN 4 Ave, Kigali P.O. Box 655, Rwanda; 3Department of Health Workforce Development, Ministry of Health, KN 3 Rd, Kigali P.O. Box 84, Rwanda; 4Center for Equity in Global Surgery, University of Global Health Equity (UGHE), Kigali Heights, KG 7 Ave, Kigali P.O. Box 6955, Rwanda; etwizeyimana@ughe.org; 5Emergency Department, Nyarugenge District Hospital, KN 247 St, Kigali P.O. Box 84, Rwanda; nsdidierdesha@gmail.com

**Keywords:** surge capacity, mass casualty incidents, resource-limited settings, East Africa: advanced trauma life support care

## Abstract

Surge capacity is the ability to manage sudden patient influxes beyond routine levels and can be evaluated using the 4S Framework: staff, stuff, system, and space. While low-resource settings like Rwanda face frequent mass casualty incidents (MCIs), most surge capacity research comes from high-resource settings and lacks generalisability. This study assessed Rwanda’s hospital surge capacity using a cross-sectional survey of emergency and surgical departments in all referral hospitals. Descriptive statistics, *t*-tests, Fisher’s exact test, ANOVA, and linear mixed-model regression were used to analyze responses. Of the 39 invited participants, 32 (82%) responded. On average, respondents believed that they could manage 13 MCI patients (95% CI: 10–16) while maintaining routine care, with significant differences between tertiary and secondary hospitals (11 vs. 22; *p* = 0.016). The intra-class correlation was poor for most variables except for CT availability and ICU beds. Surge capacity perception did not vary significantly by professional category, though less senior staff reported higher capacity. Significantly higher capacity was reported by those with continuous access to imaging (*p* < 0.01). Despite limited resources, Rwandan hospitals appear able to manage small to moderate MCIs. For larger incidents, patient distribution across facilities is recommended, with critical cases prioritized for tertiary hospitals.

## 1. Introduction

Surge capacity is defined as “the ability to manage a sudden, unexpected increase in patient volume that would otherwise severely challenge or exceed the current capacity of the health care system” [1]. The 4S framework categorises surge capacity into staff (trained personnel), stuff (comprehensive supplies and equipment), system (integrated policies and procedures), and space (facilities) [2,3]. Previous studies have assessed surge capacity through various methods, including mathematical models based on the number of manageable casualties after recruiting all possible assets [4], the influx of patients, expected resource use [5], or patient time in the emergency department [6].

Low- and middle-income countries (LMICs) carry the brunt of the global injury burden [7]. In low-resource settings, mass-casualty incidents (MCI) often overwhelm health systems, necessitating surge capacity to minimise avoidable patient morbidity and mortality and to reduce health system strain. Despite limited academic literature, LMICs are likely to be more severely impacted by MCIs than high-income countries due to higher societal vulnerabilities and less robust health systems. This hypothesis is supported by retrospective studies of MCIs in Malawi, Kenya, and Haiti, which report frequent MCIs, primarily from road traffic accidents [8,9,10], although underreporting is likely. Current literature on surge capacity originates primarily from high-resource settings, with uncertain generalisability to LMICs. Hence, studies from LMICs, where chronic resource shortage is common, are needed to make contextually appropriate policy recommendations. Additionally, such studies can test the hypothesis that low-resource settings may paradoxically be better placed than high-resource settings to manage the additional acute resource shortage that occurs during MCIs.

Rwanda is a low-income country in East Africa, where injuries, particularly due to road traffic accidents, remain a major contributor to morbidity and mortality [11,12]. Previous assessments of Rwandan emergency and surgical capacity, such as the comprehensive baseline assessment of the Rwandan National, Surgical, Obstetric and Anaesthesia Plan in 2018, have examined material and human resources [11,13], but have not reviewed systems, structures, or the dynamicity of these elements in relation to “trauma surges”. Hence, this study aims to assess the MCI surge capacity in Rwanda through a survey based on the 4S framework.

## 2. Materials and Methods

### 2.1. Definitions

“Intermediate care” was defined as a high-dependency unit (HDU) equipped with both human and material resources to support critically ill patients not requiring a ventilator or intensive care treatment. “Intensive care unit” (ICU) was defined as a critical care unit offering ventilator treatment. A “surgical team” was defined as any team constellation that made surgery possible, and this was up to the participants to interpret within their specific hospital context.

### 2.2. Study Location and the Rwandan Health System

In Rwanda, patients typically seek care at their district hospital, from where they are referred to tertiary or secondary hospitals (in this study called “referral hospitals”). All Rwandan referral hospitals, based on the designation system at the study’s conception (2023), were invited to this study. Since then, the designation system has been amended, and the included hospitals have been relabelled accordingly as level 1 hospitals, level 2 hospitals, and provincial hospitals. Hence, in this study, level 1 hospitals are considered “tertiary hospitals” and level 2 and provincial hospitals as “secondary hospitals”.

### 2.3. Surgical Capacity in Rwanda

In 2018, the Rwandan National Surgical, Obstetric, and Anaesthesia Plan (NSOAP) was launched, with goals and a strategic framework for 2024 [14]. Although there are no metrics or goals relating specifically to mass-casualty management or surge capacity, the NSOAP included a baseline assessment detailing surgical access, surgical volume, surgical specialist provider density, perioperative mortality rate, and impoverishing and catastrophic expenditure [15]. In April 2023, surveys were conducted at sixty Rwandan hospitals to evaluate the NSOAP progress [16]. These surveys indicated an increase in surgical activity, although the number of surgeries performed per 100,000 people was only half of the target (1788 surgeries, compared with the baseline of 786 and the target of 3000). Although the number of surgical specialist physicians improved threefold (from 119 to 416, equating to three surgical specialists per 100,000 people), it fell short of the goal of 16 surgeons, obstetricians, and anaesthesiologists per 100,000 people. The report also indicated shortcomings in imaging accessibility and failure to implement the governance structures suggested by the NSOAP. There is no mention of surge capacity or disaster preparedness in the NSOAP, which has been described as a missed opportunity [17].

### 2.4. Survey Design

This cross-sectional survey assessed Rwandan MCI surge capacity using the 4S Framework, focusing on emergency department (ED) and operating room (OR) capacity (Section A.1, Table A1). The survey was informed by a previous qualitative study, indicating capacity-limiting factors in all 4S dimensions (Mugabo E*, Velin L*, Trauma Care Providers’ Perceptions of Surge Capacity in Mass-Casualty Incidents in Rwanda—A Qualitative Study, unpublished [18]). Demographic information regarding the respondents’ hospital and work roles was collected. Participants were instructed to reflect on the events of the past week to reduce the risk of recall bias. The primary outcome variable was the number of MCI patients perceived could be managed without compromising care quality. Branching logic was used to tailor department-specific questions. The survey was pilot tested by five randomly selected chief nurses in referral hospitals in the emergency and surgery departments, and amended based on the testers’ feedback.

### 2.5. Study Participants

Purposive sampling was used to target participants, thought to be most knowledgeable on the matters. The inclusion criteria set before study commencement included chief nurses (matrons) and physician department heads from emergency and surgery departments (acute care surgery department when possible) in all 12 Rwandan referral hospitals, based on the designation system at the time of study conception. This included secondary (level 2 teaching hospitals or provincial hospitals) and tertiary hospitals (level 1 teaching hospitals). For secondary hospitals, a Director of Medical and Allied Health Services (DMAHS; a physician) may be responsible for all departments instead of a specific physician head of department, and a “unit manager” (chief nurse) is responsible for the day-to-day work in the department. Two hospitals had a DMAHS head. One hospital had a transition of the ED head during the study period, and for this hospital, both department heads were invited. Hence, the study population consisted of 47 potential respondents. No exclusion criteria were applied.

### 2.6. Survey Distribution

Data were collected from 24 October 2024 to 24 January 2025. Potential participants were informed about the study and invited to participate through WhatsApp, with a link to the REDCap (Vanderbilt University, Nashville, TN, USA) survey [19]. Consent was presumed if partaking in the survey. Permission to contact potential participants was sought from the DMAHS, who, after approval, shared the contact information of the participants.

### 2.7. Data Analysis

Descriptive statistics (frequencies, percentages, mean and standard deviation) were used for normally distributed data (according to the Shapiro–Wilk test). Non-normal data were log10-transformed (+0.01 for zero values) and back-transformed using exponentiation. Back-transformed values were analysed descriptively, including geometric mean (exponentiated log10-value, hereafter referred to as mean) and 95% confidence intervals (CI). Transformed variables were used for the two-sample independent *t*-test (for numeric variables) comparing tertiary and secondary hospitals. Fisher’s exact test was used for a comparative analysis of categorical (dichotomous) variables.

One-way analysis of variance (ANOVA) was used to assess intra-class correlation (ICC) between the respondents from the same hospital for numeric variables. The log-transformed version was used for the dependent variable since the ANOVA method presumes a normal distribution for the outcome variable. ICC-coefficients less than 0.50 were considered as poor correlation [20].

A mixed-model regression was used to assess the role of confounders. Linear mixed-model regression is an appropriate method when data is nested, i.e., when some data points belong to a group (in this case, respondents from the same hospital). In the mixed-model analysis, the log-transformed number of manageable patients was the dependent (outcome) variable (Section A.2, Table A2). The hospital was treated as a fixed-effect predictor. Potential confounders were treated as random intercepts, including professional role (chief nurse, general practitioner or specialist physician), imaging access (dichotomised to yes/no, with “yes” representing 24/7 access to x-ray and computer tomography (CT)), access to critical resources (dichotomised to yes/no, with “yes” representing continuous access to monitors, oxygen, pulse oximetry, facility O-negative blood and/or an on-site blood bank) and access to ICU (dichotomised to yes/no, with “yes” indicating the existence of an ICU in the hospital). *p*-values, back-transformed regression coefficients, and back-transformed 95% CI were presented. The predicted number of manageable MCI patients was presented with a full and an empty model, where logarithmised coefficients were summed and then back-transformed.

For all analyses, two-sided *p*-values < 0.05 were considered statistically significant. Respondents were contacted to re-answer the question or verify the answer in the case of outliers or missing data. There was no missing data at the stage of analysis.

### 2.8. Ethics

The study received approval from the Rwanda National Research Ethics Committee on 2 April 2024 (76/RNEC/2024). Participants were informed about the study procedures and the voluntary nature of participation; consent was presumed if participating in the study.

## 3. Results

### 3.1. Hospital and Responder Characteristics

Two hospitals (one private and one public) were not included due to a lack of institutional permission, resulting in 8 potential participants not receiving a study invitation. Of 47 potential study participants, 39 were thus contacted, of which 32 (68%) agreed to participate and were included. Hence, the response rate amongst the ten included hospitals was 82%. Amongst the respondents, 53% (*n* = 17) were chief nurses and 15 (47%) were physicians (of which eight (25%) were general practitioners and seven (22%) were specialist physicians). One declined participation, and five (13%) did not complete the survey despite multiple reminders.

The hospitals surveyed varied in geographic location, size, and catchment population (Table 1).

Respondents perceived that they could receive a mean of 13 MCI patients (95% CI: 10–16) simultaneously while maintaining regular care standards. This varied significantly between tertiary (mean 22, 95% CI: 10–34) and secondary hospitals (mean 11, 95% CI: 8–14) when comparing individual answers (*p* = 0.02). Surge capacity metrics were generally higher in tertiary compared with secondary hospitals, except for the presence of O-negative blood (so-called “emergency blood” that can be transfused without prior cross-matching) and HDU-capacity, which were non-significantly higher in secondary hospitals (Figure 1; Section A.3).

### 3.2. Descriptive Analysis of the 4S Domains

#### 3.2.1. Staff

Amongst ED staff, 17 (94%) reported having an ED physician available 24/7, and 12 (71%) reported having an ED specialist available on remote on-call duty.

Of the 59% (*n* = 19) reporting ICU capacity, 13 (68%) reported an ICU-physician being continuously available, and 16 (84%) reported an ICU-physician on remote on-call duty. ICU staff capacity could be increased during an MCI in 16/19 (84%), but only rarely with specialist nurses and physicians (*n* = 5, 26%).

Amongst the six physicians from surgery departments, the mean number of surgical teams that could be assembled within 0.5-, 2-, and 8 h was 2 (SD 1), 2 (SD 3), and 3 (SD 3), respectively.

#### 3.2.2. Stuff 

The mean number of CT machines accessible within one hour was 0.9 (95% CI: 0.59–1.1). The CT machine(s) were operational and available when needed in 53% (*n* = 17) of cases, sometimes in 13% (*n* = 4), rarely in 6% (*n* = 2), and never in 28% (*n* = 9). The mean number of X-ray machines available within one hour was 2 (95% CI 1–2). In 69% (*n* = 22) of the cases, the X-rays were always available and operational when needed, whereas six respondents (19%) reported it to be sometimes available, and two (6%) as rarely or never available, respectively.

Amongst critical resources, oxygen and blood pressure monitors were reported available 100% of the time, whereas pulse oximetry was available 94% (*n* = 30) of the time. Less than one-third (*n* = 8, 25%) of respondents had O-negative blood in the ED. Among those lacking stored blood in the ED, the mean time to emergency blood delivery was 5 h (95% CI: 0.3–9). This difference was non-significant between tertiary and secondary hospitals (secondary: mean 5 h, tertiary: 3 h, *p* = 0.7) and work positions (nurses: 8 h, physicians: 1 h, *p* = 0.1).

#### 3.2.3. Systems 

Approximately half (*n* = 17, 53%) of the respondents reported having a hospital MCI plan, whereas five (16%) were uncertain. This was significantly different when comparing hospital level, with 100% (*n* = 6/6) of all tertiary respondents reporting having an MCI plan, compared with 42% (*n* = 11/26) of secondary respondents (*p* = 0.04). All ED respondents stated that they had a triage system (*n* = 18), and five (28%) reported having a specific MCI triage system.

Three-quarters of the respondents (*n* = 24) reported having a standardised system to call in additional staff if needed, whereas one respondent did not know (3%). Twelve (39%) reported having a specific storage room with MCI supplies and equipment. Nearly two-thirds had an on-site blood bank (*n* = 20, 63%).

#### 3.2.4. Space 

The mean number of ORs available per hospital was 3 (CI 1–5), with three respondents reporting zero ORs. The difference in OR availability was statistically significant when comparing tertiary and secondary hospitals (mean 9 (CI 5–13) vs. 2 (0.5–3), *p* = 0.01). Amongst the 18 respondents reporting ICU bed availability, the mean number of ICU beds was 7 (CI 5–10), differing significantly between tertiary and secondary hospitals (mean 12 vs. 5, *p* = 0.03). The mean number of HDU beds reported was 9 (CI 5–13), and three respondents reported zero HDU beds.

### 3.3. Intra-Class Comparison

The intra-class correlation was poor for all numeric variables, except for the number of CT machines and the number of ICU beds (Table 2).

### 3.4. Regression Analysis

The perceived number of manageable MCI patients varied non-significantly between the professional categories, with nurses describing the highest capacity (9 patients), followed by general practitioners (8 patients, *p* = 0.5) and specialist physicians (5 patients, *p* = 0.07) assuming an empty regression model (no ICU availability, no imaging capacity, and critical resource shortage), Table 3. This negative impact of professional rank is demonstrated by the back-transformed coefficient being <1. Assuming a full model (ICU availability, imaging capacity, and no resource shortage), the perceived number of manageable patients was 18 for nurses, 15 for general practitioners, and 11 for specialist physicians. The perceived surge capacity was significantly higher amongst respondents reporting continuous access to imaging (with an increased capacity of three patients if having imaging capacity, *p* < 0.01).

The availability of critical resources also had a relative negative impact on the perceived number of manageable patients; however, this was not statistically significant. The impact of having an ICU was non-significant (coefficient 0.02, *p* = 0.9).

## 4. Discussion

This study is, to our knowledge, the first to assess surge capacity in a low-resource setting from trauma care providers’ perspectives. With a perceived mean capacity to manage 13 MCI-patients arriving simultaneously, without compromising care standards, Rwandan referral hospitals appear capable of managing small- to moderate-sized MCIs. However, there are wide disparities in the perceived surge capacity, with significantly higher capacity perceived in tertiary compared to secondary hospitals.

In our previous qualitative study, structures to prevent disorganisation were seen to mitigate challenging circumstances during MCIs and to reduce the risk of patient harm (Mugabo E*, Velin L* [18]). This includes surge capacity plans. However, the uptake of MCI plans appears low in Rwanda, especially in secondary facilities. Similar findings have been reported in Ethiopia, where only 17% of nurses in six regional referral hospitals (*n* = 17) reported having a disaster plan [21], and more than half were unaware of whether a disaster plan existed. This can be compared with a South African study from 2011 reporting that 93% of the 27 public hospitals studied (of 41 public hospitals in total) had a hospital disaster plan [22]. However, limited awareness of the disaster plans, similarly demonstrated in this study, and plans not being readily available in the hospital, were also reported in South Africa [22] and are well-known problems also in high-income countries [23].

A shared mental model is crucial for surge capacity. The varied responses on perceived capacity, reflected in poor intra-class correlation, highlight a lack of common understanding. For example, 16% of respondents (*n* = 5) were unsure if their hospital had an MCI plan, whilst other respondents from these hospitals answered yes or no. Of note, all uncertain responses were from nurses from secondary hospitals. A similar nurse-physician divide emerged in estimates of manageable patient numbers, with increasing seniority correlating with more conservative estimates. A previous study of Swedish ED nurses similarly found an overestimation of their disaster competency and increasing competency in disaster management with increasing nurse seniority [24]. Assuming that the responses from the specialist physicians are the gold standard, this may reflect junior team members having a less comprehensive understanding of trauma care and less MCI experience. These findings underscore the importance of team training and specific disaster management training for nurses. While all team members contribute unique and valuable perspectives to trauma care, the findings also highlight the need to involve doctors and, if possible, senior physicians as the main clinical experts in disaster planning.

Trauma system components such as imaging and intensive-care capacity are generally viewed as critical to surge capacity. Amongst the 4S domains, access to imaging was the only statistically significant factor impacting perceived surge capacity. This study indicates that 1 h CT access was unavailable to most hospitals, and that even when in place, they were only perceived to be continuously available when needed by half of the respondents. This aligns with findings from Tanzania, where no regional hospitals had CT access [25]. Although x-ray access in this study was better, nearly double the Tanzania average of one x-ray machine per hospital [25], inoperability issues commonly hindered its usage, similar to other sub-Saharan African countries [26]. Since the launch of the Rwandan NSOAP in 2018, progress has been made regarding C-arm, ultrasound, and teleradiology [16]. However, no goals or strategy for CT access were defined in the NSOAP [14], despite its central role in trauma management, highlighting a need to prioritise imaging in future efforts to develop surge capacity. In contrast, there was no significant association between ICU capacity and perceived surge capacity. Although a true association may exist, but not be captured in this sample, the existence of a small ICU may not equate to a significantly higher capacity to manage critically ill patients. In a nationwide survey of critical care capacity in Ethiopia, only 63% of ICU beds had ventilators [27], whereas a study of all tertiary referral hospitals in Tanzania, albeit from 2014, found that none of the ICUs had an ICU-trained specialist and that only one had a critical care nurse [28]. In South African ICUs, only 25% of nurses were ICU-qualified [22]. In Rwanda, patients on ventilators often remain in the ED for several days due to overcrowded and limited capacity of ICUs, blurring the distinction in the level of care provided.

The availability of critical resources negatively influenced the number of manageable patients, although this association was non-significant. This is likely a result of the small sample and does not appear to have clinical relevance. In contrast, the literature supports the association between resource availability and surge capacity. However, the findings may also result from the fact that nearly all hospitals had access to basic monitoring, and amongst the hospitals lacking emergency blood or an on-site blood bank, there were other robust systems for rapid blood delivery. In our study, eight (25%) respondents reported in-hospital blood storage, whereas only 9% of hospitals, according to the NSOAP review in 2023, lacked blood storage [16]. Yet, the NSOAP review also described that 57% typically had only 1 to 10 units of blood products available, and 34% had more than 10 units [16]. These storages may be emptied quickly during MCIs, and hence, it must be concluded that emergency blood storage is inadequate for larger MCIs in Rwanda. According to the NSOAP, the goal is that blood should be available within two hours [14], which did not appear to be the case amongst responders lacking an on-site blood bank, reporting a mean time to emergency blood delivery of 3 h (95% CI: 0.7–7). In a subset analysis, the mean reported by physicians was 1 h (95% CI: 0.9–2), which better aligns with the NSOAP review from 2024, which reported that 94% of hospitals were within two hours of a blood bank [16], and a study of drone-delivery of blood in Rwanda reporting a mean delivery time of 50 min [29]. To conclude, blood products and basic triage tools are likely not rate-limiting factors to surge capacity in Rwanda. Still, two-hour access is an insufficient target for emergency cases, and measures to ensure immediate transfusion capability and on-site storage of larger quantities of O-negative blood are necessary.

The number of surgical teams that can be mobilised was limited in the studied hospitals, with the mean numbers ranging from 2 to 3 teams. The only respondent from a tertiary setting reported that 3, 7, and 8 teams could be mobilised within 30 min, 2 h, and 8 h, respectively, indicating a larger capacity in tertiary hospitals. A larger surgical surge capacity in tertiary hospitals is supported by the statistically significantly larger OR availability. The findings can be compared with a Swedish study, which found that an average of 3, 5, and 11 teams could be mobilised in university hospitals within 30 min, 2 h, and 8 h, respectively [30]. Amongst county hospitals, the corresponding numbers were 2, 4, and 7. The limited capacity in Rwanda to mobilise surgical teams is likely due to limited surgical staff [14,16], and limited OR availability.

### Strengths and Limitations

This study consists of self-reported data and utilizes trauma responders’ perceptions as a proxy for surge capacity. This may be subject to common biases; however, we believe that the advantages outweigh the risks, as this is real-life data that reflects the complexities of actual practice. Moreover, in contrast to previous surveys, which have addressed senior administrators [23,30], we believe that our sample choice likely depicts a truer version of the clinical realities, as frontline staff can answer questions based on experience. Discrepancies between our findings and other reports, such as blood product availability, raise questions about which data is more valid and reliable. While objective resource assessments reduce bias, staff perceptions may better reflect the standard of care over time, especially in LMICs where stock-outs and equipment failures create variability. Factors such as the availability of water sterilisation units, laundry units, and electricity, which have been documented as robust in the Rwandan NSOAP review [16], were excluded from this survey, as the respondents were not expected to have limited knowledge of these matters. Finally, in this study, we assess trauma leaders’ perceptions of surge capacity. However, further studies should assess the validity of these findings, including whether the care standards are respected when managing surges through the assessment of treatment and patient outcomes.

The survey was designed for this study and should be validated before wider adoption. While the World Health Organisation National Health Sector Emergency Preparedness and Response tool [31] was an option, our survey was deemed more clinically relevant. Further detail would have been helpful; however, the survey was condensed since the respondents had clinical duties, limiting their time. Validated surgical assessment tools [32,33,34] were deemed unsuitable based on their sole focus on surgical capacity and not surge capacity dynamics. This study investigated the clinical services of most importance in the first 24 h post-MCI; however, surge capacity must also address institutional aspects, pre-hospital services, and community-based surge capacity. Yet, for small to moderate MCIs, which are most common in Rwanda, ED, surgery, ICU, and emergency medical services (EMS) are likely most crucial. A follow-up study will assess the EMS perspective.

Intra-hospital response variability may be seen as a methodological concern. Social desirability and reporting bias, including exaggeration of the hospital capacity, must be considered, making the results a best-case scenario. This variability likely reflects true differences in MCI capacity perceptions and understanding among staff categories. Intra-class correlation analysis helped quantify these differences and is a study strength. For categorical data, descriptive comparisons were preferred over formal agreement (e.g., Fleiss’ and Cohen’s Kappa), due to the branching logic with varying rater numbers.

The generalisability of the results to similar low-resource settings may be discussed. For practical purposes of surge capacity assessment to develop disaster plans, local assessments must be conducted. However, we believe that the overarching patterns described are likely to be accurate in comparable low-resource settings with a high trauma burden.

## 5. Conclusions

Despite limited formal routines and shortages of human and material resources, the perceived surge capacity in Rwanda is adequate to manage small to moderate MCIs. For large events, patient distribution to multiple facilities should be considered, with critical cases directed to tertiary settings, where surge capacity is better. Further work is needed to ensure a shared mental model amongst staff, including increased awareness of existing MCI plans. Future studies should validate surge capacity through treatment and outcome metrics to ensure that care quality is maintained during MCIs.

## Figures and Tables

**Figure 1 ijerph-22-01559-f001:**
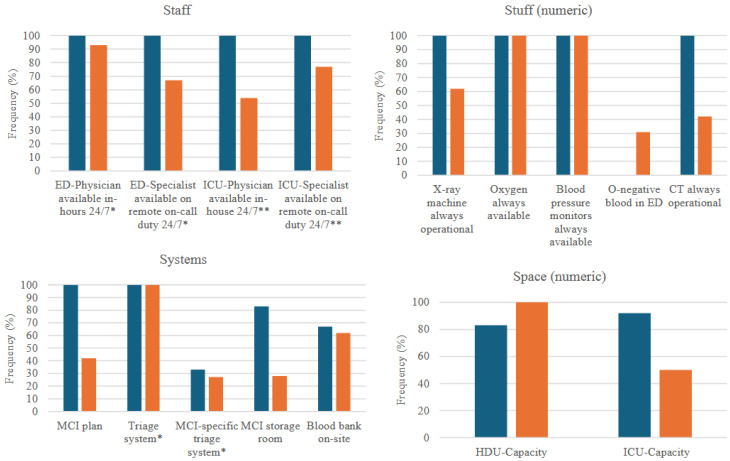
Data presented as frequencies (categorical variables), with 6 respondents from tertiary hospitals and 26 respondents from secondary hospitals. * Branching logic with questions only asked to ED staff (*n* = 2 and 15, respectively). ** Branching logic with questions only asked to those reporting ICU capacity (*n* = 6 and 13, respectively). ICU = Intensive Care Unit; CT = Computer Tomography; ED = Emergency Department, MCI = Mass-Casualty Incident; HDU = High-Dependency Unit.

**Table 1 ijerph-22-01559-t001:** Hospital characteristics.

Hospital Name	Province	Hospital Classification	Catchment Population	Number of Hospital Beds *
University Teaching Hospital of Butare (CHUB)	Southern	Level 1 Teaching Hospital	3.7 million	500
University Teaching Hospital of Kigali (CHUK)	Kigali	Level 1 Teaching Hospital	6.2 million	519
Kibungo Level 2 Teaching Hospital	Eastern	Level 2 Teaching Hospital	410,000	293
Ruhengeri Level 2 Teaching Hospital	Northern	Level 2 Teaching Hospital	406,557	328
Kibuye Level 2 Teaching Hospital	Western	Level 2 Teaching Hospital	197,491	206
Butaro Level Two Teaching Hospital	Northern	Level 2 Teaching Hospital	350,000	250
Bushenge Provincial Hospital	Western	Provincial Hospital	170,000	212
Ruhango Provincial Hospital	Southern	Provincial Hospital	228,992	192
Rwamagana Level Two Teaching Hospital	Eastern	Level 2 Teaching Hospital	369,671	242
Kinihira Provincial Hospital	Northern	Provincial Hospital	145,249	320

* Data on the number of hospital beds was provided by the Department of Clinical and Public Health Services within the Rwandan Ministry of Health (2025), and the catchment population was retrieved from the hospital websites (March 2025).

**Table 2 ijerph-22-01559-t002:** Intra-class comparison with ICC-coefficients, 95% CI, and the interpretation.

Variable	ICC (95% CI)	Interpretation
MCI patient number capacity	0.2 (0.0–0.6)	Poor reliability
Number of ORs	0.0 (000–1.0)	Poor reliability
Number of CTs	0.8 (0.5–1.0)	Moderate/good reliability
Number of X-ray machines	0.1 (0–0.5)	Poor reliability
Time to blood delivery	0.4 (0–0.9)	Poor reliability
Number of ICU beds	0.8 (0.5–1)	Good reliability
Number of HDU beds	0.2 (0.0–0.6)	Poor reliability

**Table 3 ijerph-22-01559-t003:** Final regression model using linear mixed-model regression analysis, with log-transformed coefficients and 95% CI, and *p*-values.

Model Parameter	Coefficient (95% CI)	*p*-Value
Constant	9 (5–18)	
ICU availability	1 (0.6–2)	0.8
Imaging availability	3 (2–5)	<0.01
Critical resource availability	0.7 (0.4–1)	0.09
Professional role (nurse = reference category) -General practitioner -Specialist physician	0.8 (0.5–1) 0.6 (0.4–1)	0.5 0.07

## Data Availability

Study data can be shared upon reasonable request.

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
