# Peer review of "Quantitative Assessment of Surge Capacity in Rwandan Trauma Hospitals: A Survey Using the 4S Framework"

_ijerph, 2025, doi:10.3390/ijerph22101559_

Round 1
Reviewer 1 Report
Comments and Suggestions for Authors
The paper aims to quantify hospital “surge capacity” for trauma in Rwanda using the 4S framework (staff, space, systems, and supplies) through a cross-sectional survey of emergency and surgical leaders in referral hospitals. Its main contribution is a practical, frontline perspective estimate of how many simultaneous MCI patients hospitals believe they can handle without disrupting routine care and which capacity components are most associated with that perception. Briefly, 32 respondents from 10 hospitals reported an average perceived capacity of 13 concurrent MCI patients, with higher estimates in tertiary hospitals compared to secondary hospitals. Access to imaging, especially CT within one hour, was the only factor significantly linked to higher perceived capacity in mixed-effects models. The study also reveals poor intraclass agreement across many variables, indicating a patchy “shared mental model” within hospitals, and notes the low awareness or use of formal MCI plans, particularly outside tertiary centers. The authors’ bottom line is pragmatic: small-to-moderate MCIs seem manageable, while larger events likely require inter-facility distribution with triage of critical cases to tertiary hospitals.
Regarding design and conduct, this is a standard, service-oriented cross-sectional survey with reasonable face validity for its purpose. The sampling frame, which includes chief nurses/matrons and physician department heads in ED and surgery across all referral hospitals as identified during study planning, is appropriate for a perception survey. The timeline (Oct 24, 2024–Jan 24, 2025) is clearly outlined. Recruitment through DMAHS contacts and delivering a REDCap link via WhatsApp are practical methods within the context; consent procedures (assumed via completion) received approval from the national ethics committee. I commend the team for pilot testing the instrument with five chief nurses, customizing branching logic based on departmental roles, and trying to reduce recall bias by asking respondents to “reflect on the past week.” All of these steps are suitable for a quick, operational snapshot.
That said, several aspects could be reported more clearly. First, regarding inclusion and exclusion: the paper states that all referral hospitals were selected based on the then-current designation; however, two (one public, one private) were ultimately excluded due to a lack of institutional permission.
At lines 169–170, a sentence is repeated (“Of 47 potential study participants, 39 were thus contacted, of which 32 (68%) agreed…”).
Regarding the statistical approach, the pipeline primarily includes: Shapiro–Wilk to assess normality, log10 transformation with a +0.01 offset for zeros, back-transformed descriptive statistics, two-sample t-tests on transformed metrics, Fisher’s exact tests for dichotomous variables, one-way ANOVA to compute ICC for numerical variables, and a linear mixed model for the main outcome with hospital clustering. I have some suggestions:
- The mixed-model description states, “hospital was the independent variable,” which suggests that hospital was treated as a fixed-effect predictor. For clustered survey data, you almost certainly want to treat hospital as a random intercept (or random intercept/slope where appropriate). The authors should clarify their model structure, including whether they used random or fixed effects, the estimation method, and the covariance structure. They should also provide the intraclass correlation for the dependent variable from the mixed model, not just from the ANOVA-based ICC. If hospital was included as a fixed effect instead of a random effect, the inferences about imaging/ICU/confounders could be biased by unmodeled within-hospital correlation. Please specify the exact R (or other software) syntax or at least clarify the modeling choices.
- Several confidence intervals and means in Table 2 are not credible as printed. For example, “Time to blood delivery (hrs) 3.2 (−0.72–7.1)” cannot have a negative lower bound; similarly, “Number of X-ray machines” for secondary hospitals shows “1.9 (1.2–25)” with an implausible upper CI of 25 machines. These appear to be formatting or back-transformation/reporting errors. The CT count line reads “1.7 0.51 0.01,” which seems to have lost its CI and p-value labels. A careful table audit is needed.
The discussion is generally well-balanced. I appreciated the comparison to Ethiopia, Tanzania, and South Africa, as well as the honest acknowledgment that systems and awareness (such as MCI plans and call-in procedures) are just as important as hardware. The segment about imaging is the most compelling and actionable: in this dataset, 1-hour CT access is limited, and even when available, continuous access is uncertain, while imaging access shows the strongest statistical link in the mixed model. This connection seems clinically appropriate and relevant for revisions to NSOAP. The section on blood products is thoughtful but could be improved. The paper compares survey perceptions (e.g., only 25% reported O-negative in the ED; long delivery times without an on-site blood bank) with NSOAP review statistics, and the authors interpret discrepancies, referencing Rwanda’s drone blood delivery literature.
The bibliography is mostly relevant and current.
In the survey design, an “Appendix 1” is mentioned, but it is not included.
Overall, I believe this is a practical, experience-based piece that provides policymakers and hospital leaders with a clear, Rwanda-specific starting point for enhancing trauma surge capacity. However, some corrections are necessary.
Reviewer 2 Report
Comments and Suggestions for Authors
1. Introduction:
- Authors needs to make the knowledge gap more explicit (e.g., what hasn’t been studied in Rwanda or LMICs that this survey uniquely addresses).
- Authors clarify how this study complements or fills gaps in national policy (NSOAP).
- A stronger closing statement in the introduction to frame the study’s contribution beyond descriptive assessment.
2. Justify more clearly why perceptions (rather than objective measures) are an acceptable proxy for surge capacity.
3. Discuss representativeness: the response rate was 82% of invited, but only 32 respondents in total—highlight the limits of generalizability.
4. Provide rationale for purposive sampling of department heads and nurses versus inclusion of broader staff.
5. Clarify missing data handling beyond “respondents were contacted.”
6. Explain more on how recall bias was minimized (one-week recall is mentioned, but reinforce why this timeframe was chosen).
7. Strengthen justification for using mixed-model regression (e.g., nested hospital-level clustering) for clarity to readers less familiar with the method.
8. Results
- The narrative could highlight key takeaways first, before detailed numbers.
- Intra-class correlation results could be more clearly linked to implications (e.g., what does poor reliability mean for practice?).
9. Conclusion
- Link findings more explicitly to policy recommendations.
- Add a statement about the need for validation of survey tool before wide adoption.
10. Figures and Tables
- Too many numbers in tables may overwhelm readers. Use more visual summaries (bar graphs, flow diagram of survey participation, or side-by-side plots for tertiary vs. secondary hospitals).
- Ensure consistency in reporting (e.g., decimals, CI format).
- Some long tables (like Table 2) could be split or placed in appendices to streamline the main text.
Author Response
Thank you very much for taking the time to review this manuscript. Please find the detailed point-by-point responses below (in red) and the corresponding revisions/corrections in track changes in the re-submitted files.
Lotta Velin on behalf of all co-authors
Reviewer 2
Introduction:
- Authors needs to make the knowledge gap more explicit (e.g., what hasn’t been studied in Rwanda or LMICs that this survey uniquely addresses).
Response: This section has been reworded to more explicitly highlight the knowledge gap, which now reads: “Current literature on surge capacity originates primarily from high-resource settings, with uncertain generalisability to LMICs. Hence, studies from LMICs, where chronic resource shortage is common, are needed to make contextually appropriate policy recommendations. Additionally, such studies can test the hypothesis that low-resource settings may paradoxically be better placed than high-resource settings o manage the additional acute resource shortage that occurs during MCIs.”
- Authors clarify how this study complements or fills gaps in national policy (NSOAP).
Response: Due to word count restrictions, we feel unable to expand further upon this point. However, in the introduction, a link to the NSOAP is made, and the complementary nature of this study is illustrated (the dynamicity required in a surge): “Previous assessments of Rwandan emergency and surgical capacity, such as the comprehensive baseline assessment of the Rwandan National, Surgical, Obstetric and Anaesthesia Plan in 2018, have examined material and human resources 11,13, but have not reviewed systems, structures, or the dynamicity of these elements in relation to “trauma surges”. Hence, this study aims to assess the MCI surge capacity in Rwanda through a survey based on the 4S framework.
- A stronger closing statement in the introduction to frame the study’s contribution beyond descriptive assessment.
Response: Thank you for this suggestion. We hope and believe that this issue has been addressed through the changes made in response to the first point. Hence, see the previous response (first point in “Introduction”).
- Justify more clearly why perceptions (rather than objective measures) are an acceptable proxy for surge capacity.
Response: We thank you for this valid point. We already discussed this in part in the strengths and limitations section, and hence we have decided to add a statement with justification for this in this section, to ensure flow and readability. The section now reads: “This study consists of self-reported data and utilizes trauma responders’ perceptions as a proxy for surge capacity. This may be subject to common biases; however, we believe that the advantages outweigh the risks, as this is real-life data that reflects the complexities of actual practice. Moreover, in contrast to previous surveys, which have addressed senior administrators 22,29, we believe that our sample choice likely depicts a truer version of the clinical realities, as frontline staff can answer questions based on experience.” It is followed by comparison with “objective” measures from for example the NSOAP, and the advantages and disadvantages of the different data types is further discussed here.
- Discuss representativeness: the response rate was 82% of invited, but only 32 respondents in total—highlight the limits of generalizability.
Response: The generalisability of the study may be discussed on different scales. We address the point of generalisability in the strengths and limitations, where we suggest that, although there may be some “lessons to be learnt” for similar low-resource contexts, direct extrapolation to other contexts may have limited practical utility, as surge capacity should be informed by local data. We believe that this is already described, and have hence not made any further changes. Regarding the sample strategy, we expand on this point in the next comment.
- Provide rationale for purposive sampling of department heads and nurses versus inclusion of broader staff.
Response: Due to word limit restrictions, we feel unable to expand significantly on this, but we decided to target chief nurses and physician head of departments since they have responsibility for the clinical work in the departments, including ensuring disaster preparedness and surge capacity. Hence, we decided to target these as they were thought to be the most knowledgeable about the matters. Estimates for for example, time needed to blood delivery could have been significantly skewed if staff without knowledge on the matters made random estimations, not based on significant experience. We have adjusted the section that now reads: “Purposive sampling was used to target participants, thought to be most knowledgeable on the matters. The inclusion criteria set before study commencement included chief nurses (matrons) and physician department heads from emergency and surgery departments (acute care surgery department when possible)…”
- Clarify missing data handling beyond “respondents were contacted.”
Response: Thank you for this comment, which highlights the lack of clarity regarding missing data. In fact, upon contact with respondents, we were able to fill in the data gaps, and hence, there was no further analysis of missing data. We have clarified this section, which now reads: “Respondents were contacted to re-answer the question or verify the answer in the case of outlier or missing data. There was no missing data at the stage of analysis”.
- Explain more on how recall bias was minimized (one-week recall is mentioned, but reinforce why this timeframe was chosen).
Response: One-week recall is a relatively common timeframe within research, arguably even the norm, and was thought to be a practically useful timeframe. Hence, we do not believe that this needs further elaboration in the manuscript.
- Strengthen justification for using mixed-model regression (e.g., nested hospital-level clustering) for clarity to readers less familiar with the method.
Response: We have reworded this method section to enhance transparency and additionally added a table in the supplementary material, so that readers can more easily follow the regression process. This section now reads: “A mixed-model regression was used to assess the role of confounders. Linear mixed-model regression is an appropriate method when data is nested, i.e., some data points belong to a group (in this case, respondents from the same hospital). In the mixed-model analysis, the log-transformed number of manageable patients was the dependent (outcome) variable. Hospital was treated as a fixed-effect predictor. Potential confounders were treated as random intercepts, including professional role…” We hope and believe that the method choice is now clearer.
- Results
- The narrative could highlight key takeaways first, before detailed numbers.
Response: We believe that the takeaways should be addressed in the discussion section, and hence prefer to keep the results as it is, and have decided to not make any changes regarding this point.
- Intra-class correlation results could be more clearly linked to implications (e.g., what does poor reliability mean for practice?).
Response: We discuss the implications of the poor intra-class correlation on page 9, lines 492-506 and page 10, lines 596.602.
- Conclusion
- Link findings more explicitly to policy recommendations.
Response: We believe that this study alone does not have enough data to provide extensive policy recommendations. However, we do make the following suggestion in the conclusion: “For large events, patient distribution to multiple facilities should be considered, with critical cases directed to tertiary settings, where surge capacity is better.” This study is a part of a PhD thesis (cumulative dissertation) consisting of in total four manuscripts assessing MCI epidemiology and surge capacity in Rwanda, and we believe that they together provide a better basis for policy recommendations. Hence, further policy recommendations will be presented in the published dissertation, which will also be made available (open-access) online.
- Add a statement about the need for validation of survey tool before wide adoption.
Response: Thank you for this suggestion. We have added this point on page 11, line 584, which now reads: “The survey was designed for this study and should be validated before wider adoption.”
- Figures and Tables
- Too many numbers in tables may overwhelm readers. Use more visual summaries (bar graphs, flow diagram of survey participation, or side-by-side plots for tertiary vs. secondary hospitals).
Response: Thank you for this comment. We have converted the categorical variables in Table 2 to a bar chart, as suggested. To ensure that no data is lost, we have added the original Table 2 as supplementary material.
- Ensure consistency in reporting (e.g., decimals, CI format).
Response: Thank you for this comment. We have reviewed the manuscript thoroughly to ensure that decimals are now consistently reported.
- Some long tables (like Table 2) could be split or placed in appendices to streamline the main text.
Response: See the comment above, in response to this suggestion.
